# How Timely Is Diagnosis of Lung Cancer? Cohort Study of Individuals with Lung Cancer Presenting in Ambulatory Care in the United States

**DOI:** 10.3390/cancers14235756

**Published:** 2022-11-23

**Authors:** Monica Zigman Suchsland, Lesleigh Kowalski, Hannah A. Burkhardt, Maria G. Prado, Larry G. Kessler, Meliha Yetisgen, Maggie A. Au, Kari A. Stephens, Farhood Farjah, Anneliese M. Schleyer, Fiona M. Walter, Richard D. Neal, Kevin Lybarger, Caroline A. Thompson, Morhaf Al Achkar, Elizabeth A. Sarma, Grace Turner, Matthew Thompson

**Affiliations:** 1Department of Family Medicine, University of Washington, Seattle, WA 98195, USA; 2Department of Biomedical Informatics and Medical Education, University of Washington, Seattle, WA 98195, USA; 3Department of Health Systems and Population Health, School of Public Health, University of Washington, Seattle, WA 98195, USA; 4Department of Surgery, University of Washington, Seattle, WA 98195, USA; 5Department of Medicine, University of Washington, Seattle, WA 98195, USA; 6Wolfson Institute of Population Health, Barts and The London School of Medicine and Dentistry, Queen Mary University of London, London E1 4NS, UK; 7The Primary Care Unit, Department of Public Health and Primary Care, University of Cambridge, Cambridge CB1 8RN, UK; 8University of Exeter Medical School, University of Exeter, Exeter EX1 2LU, UK; 9Department of Information Sciences and Technology, George Mason University, Fairfax, VA 22039, USA; 10Department of Epidemiology, Gillings School of Global Public Health, The University of North Carolina at Chapel Hill, Chapel Hill, NC 27599, USA; 11Division of Epidemiology and Biostatistics, School of Public Health, San Diego State University, San Diego, CA 92182, USA; 12Healthcare Delivery Research Program, Division of Cancer Control and Population Sciences, National Cancer Institute, Bethesda, MD 20892, USA

**Keywords:** lung cancer, diagnosis, ambulatory care, natural language processing, diagnostic intervals

## Abstract

**Simple Summary:**

Lung cancer is the most common cause of cancer related death in the US, but survival is far better when people are diagnosed at an earlier stage. There are currently no clinical quality measures that are routinely used to measure the quality or timeliness of diagnosis of lung cancer in the US. We used Natural Language Processing (NLP) to extract information on the symptoms and signs that had been recorded in the electronic medical records of patients presenting in ambulatory care over the 2 years prior to their diagnosis with lung cancer. We found that the time from the first recorded symptoms/signs associated with lung cancer to diagnosis was 570 days. The time intervals from chest CT or chest X-ray imaging to diagnosis, and from specialist consultation to diagnosis were shorter—at 43 and 72 days, respectively. Advanced techniques such as NLP can be used to extract detailed information from electronic medical records, that could potentially be used to create clinical quality measures with the goal of improving the timeliness of diagnosis of this cancer.

**Abstract:**

The diagnosis of lung cancer in ambulatory settings is often challenging due to non-specific clinical presentation, but there are currently no clinical quality measures (CQMs) in the United States used to identify areas for practice improvement in diagnosis. We describe the pre-diagnostic time intervals among a retrospective cohort of 711 patients identified with primary lung cancer from 2012–2019 from ambulatory care clinics in Seattle, Washington USA. Electronic health record data were extracted for two years prior to diagnosis, and Natural Language Processing (NLP) applied to identify symptoms/signs from free text clinical fields. Time points were defined for initial symptomatic presentation, chest imaging, specialist consultation, diagnostic confirmation, and treatment initiation. Median and interquartile ranges (IQR) were calculated for intervals spanning these time points. The mean age of the cohort was 67.3 years, 54.1% had Stage III or IV disease and the majority were diagnosed after clinical presentation (94.5%) rather than screening (5.5%). Median intervals from first recorded symptoms/signs to diagnosis was 570 days (IQR 273–691), from chest CT or chest X-ray imaging to diagnosis 43 days (IQR 11–240), specialist consultation to diagnosis 72 days (IQR 13–456), and from diagnosis to treatment initiation 7 days (IQR 0–36). Symptoms/signs associated with lung cancer can be identified over a year prior to diagnosis using NLP, highlighting the need for CQMs to improve timeliness of diagnosis.

## 1. Introduction

Lung cancer is the most common cause of cancer-related death in the United States (US) with 5-year survival rates for non-small cell lung cancer (NSCLC) and small cell lung cancer (SCLC) of 25% and 7%, respectively [1]. While early stage lung cancer can often be treated surgically, ninety percent of those diagnosed with lung cancer will die of the disease in part due to late-stage (Stages III or IV) presentation and lethality of this disease [2]. Prognosis largely reflects stage at diagnosis, with 5-year survival rates for early stage localized lung cancer of 63% and 27% for late-stage lung cancer [1].

While screening asymptomatic individuals for lung cancer using low dose computed tomography (LDCT) in high-risk patients has been recommended in the US since 2012 [3], uptake has been limited and most individuals with lung cancer are diagnosed following symptomatic presentation [4,5]. Detection of potential lung cancer among symptomatic individuals is challenging as their symptoms are difficult to distinguish from more common conditions [6,7,8]. Moreover, the time interval from symptom onset to diagnosis of lung cancer can be considerable; a study of over 48,000 patients using Medicare claims identified a median interval from symptom onset to diagnosis of 187 days (25–75% IQR 36–308 days) [9]. The long pre-diagnosis time interval for symptomatic lung cancer may be a target for quality improvement interventions with the aim of earlier detection. Notably, diagnostic errors related to lung cancer are among the most common causes of major malpractice claims in outpatient settings [10,11].

Clinical quality measures (CQM) are used to evaluate or set benchmarks of processes, outcomes, patient perceptions, or organizational structures in healthcare that equate to higher quality care and drive institutional improvements. The World Health Organization, for example, has set a goal of 90 days from symptom onset to treatment of lung cancer [12], while guidelines in Canada recommend time from presentation to diagnosis should be a maximum of four weeks [13,14]. In Denmark, clinical quality indicators have been used for more than 20 years to improve a range of outcome indicators for lung cancer [15]. However, despite the significant potential opportunities to improve care in the US [16], there are no CQMs focused on the timeliness of diagnosis of lung cancer currently used in the US.

Developing a CQM for diagnosis of lung cancer is complex; this condition includes multiple parts of the health care system (e.g., primary care, specialists, inpatient care, radiology) which would be reflected in the multiple types and sources of data needed to populate a CQM. Current gaps in defining a CQM include how to define and operationalize key timepoints in the pre-diagnosis period using electronic health record (EHR) data, as well as defining expected ranges for time intervals. The overall aim of this study was therefore to define key time points and describe intervals in the diagnostic pathway of patients with lung cancer based on a single site in the US, from initial symptomatic presentation in ambulatory care to diagnosis, using routine EHR data. We believe our findings provide valuable new information that not only describe the timeliness of the diagnostic process for lung cancer, but could also potentially be used to inform efforts to develop CQM for lung cancer in a range of health care settings

## 2. Methods

### 2.1. Study Design

A retrospective cohort of patients who received ambulatory care at UW Medicine (UWM), a health system affiliated with the University of Washington (Seattle, Washington) with a diagnosis of a first, primary lung cancer [17]. This study was approved by the University of Washington Human Subjects Division (STUDY00008248 and STUDY00013191).

### 2.2. Participants

Eligible patients were 18 years and older, with a first primary lung cancer diagnosed between 1 January 2012–31 December 2019, who had an established relationship with UWM ambulatory care, and chest computed tomography (CT) performed at UWM prior to their first recorded lung cancer diagnostic code. An ambulatory care relationship was defined as the presence of at least one visit to the following department specialties in the 24 months prior to the first recorded lung cancer diagnostic code: family medicine, internal medicine, women’s health, obstetrics and gynecology, urgent care, and/or emergency medicine. The requirements for ambulatory care relationship and chest CT were used to ensure that patients had received pre-diagnostic care and confirmatory imaging within UWM, rather than referrals from outside healthcare systems (e.g., only for specialty care).

### 2.3. Data Collection

Data were collected through the UWM enterprise-wide data warehouse (EDW), a secure central repository that integrates EHR data across UWM. The EDW was queried for patients with lung cancer identified by ICD 9 or 10 diagnostic codes during the study period. Patients with tracheal cancer, mesothelioma, Kaposi’s sarcoma, and lymphoma/leukemia histology codes were excluded. De-identified EHR data were extracted for all encounters in the 24 months prior and 6 months following the first recorded diagnostic code for lung cancer. We chose the pre-diagnosis time interval to provide data to fulfill the ambulatory care relationship (as noted above), and post-diagnosis to ensure there was complete data to verify or cross-check date of diagnosis. Extracted data included demographics (smoking status, age, sex, race, ethnicity), all ICD 9 or 10 diagnostic and Current Procedural Terminology (CPT) codes linked to encounters, as well as unstructured clinical notes for any of the above encounters. Study records from the EDW were then linked to the Seattle/Puget Sound Surveillance, Epidemiology, and End Results (SEER) Program cancer registry which provided history of previous cancers, histology, date and stage of diagnosis, and date of initial treatment.

### 2.4. Sociodemographic Variables & Comorbidity

The UWM data were used to determine age at diagnosis, sex, race and ethnicity, and smoking status (ever smoker defined as current or past smoking; never smoker defined as no history of any smoking). SEER registry provided data on health insurance and poverty which used the Census Tract Poverty Indicator to categorize individuals’ residence into categories of 10%, 15%, or 20% of people in the census tract living below the federally defined poverty line [18]. Comorbidity was calculated using the Elixhauser comorbidity index (ECI) [19]; ICD 9 and 10 diagnostic codes in the 2 years prior to lung cancer diagnosis were searched for 31 potential comorbidities. The sum and type of comorbidity were used to calculate van Walraven weighted score for each patient [20,21]. Patients who had low-dose computed tomography (LDCT) lung cancer screening within the 12 months prior to diagnosis date were identified from codes for LDCT screening linked specifically to billing codes (CPT 71271 and G0297) and/or ICD code (V76.0 [ICD-9] or Z12.2 [ICD-10]) in patients without a lung cancer diagnosis prior to that visit.

### 2.5. Time Point Definitions

Definitions of key pre-diagnosis time points were adapted from international cancer reporting standards [22,23,24]; (A) First symptomatic presentation, (B) Referral for or receipt of initial chest imaging (chest X-ray or chest CT), (C) Referral to or encounter with a specialist (i.e., Ambulatory Surgery, General Surgery, Hematology, Interventional Radiology, Medical Oncology, Neuro Oncology, Oncology, Palliative Care, Pulmonary Diagnostic Testing, Pulmonary Medicine, Radiation Oncology, Radiation Therapy, Respiratory Disease, Sarcoma, Special Procedures, Surgery, Thoracic, Thoracic Medicine, Thoracic Surgery), (D) Date of diagnosis, and (E) Date of first treatment (Appendix A). Date of first symptomatic presentation was based on the presence of symptoms or signs that have previously been identified as significantly associated with the presence of lung cancer compared to matched controls from the same population [17]. These were: finger clubbing, lymphadenopathy, cough, hemoptysis, chest crackles or wheeze, weight loss, back pain, bone pain, shortness of breath, fatigue or chest pain. We used two approaches to identify these clinical features from the EHR in the 2 years prior to diagnosis: (1) ICD9/10 codes matched to the above clinical features, (2) Application of a natural language processing (NLP) framework to extract these clinical features from the unstructured data found in the free text of clinical notes [25].

### 2.6. Lung Cancer Histology and Stage

SEER histology codes were used to categorize cancer type as small cell lung cancer (SCLC) (ICD-0 histology codes 8041-8045), non-small cell lung cancer (NSCLC) (ICD-0 histology codes 8000-8040 or 8046-9989), and other [26,27,28]. Stage variables in SEER were derived from the American Joint Committee on Cancer (AJCC) TNM Staging System, Extent of Disease (EOD), or Collaborative Stage (CS), depending on year of diagnosis. SEER staging variables were grouped into stage 0, stage I, stage II, stage III, stage IV, not applicable, stage occult, and stage unknown [29].

### 2.7. Data Analysis

Frequencies and counts were calculated for patient characteristics overall and by lung cancer stage and type. Groupwise comparisons using chi-square for categorical variables and t test for continuous variables were performed to determine significant differences. The van Walraven weighted score [20,21] was calculated using the comorbidity package in R. We calculated time intervals in days for each patient and summarized these using mean, standard deviation (SD), median, and interquartile range (IQR). Intervals calculated included: first clinical presentation to initial chest imaging (chest X-ray or chest CT) (timepoint A to B), first clinical presentation to referral/encounter with specialist (timepoint A to C), and first clinical presentation to diagnosis (timepoint A to D). Intervals were also categorized by stage (early stages I/II vs. late stages III/IV) and type of cancer (SCLC vs. NSCLC). Analyses were conducted using RStudio (Version 1.4.1106, RStudio, Inc., Boston, MA, USA) and the Statsmodels package (version 0.11.1) for Python 3.7 [30]. This study is reported in compliance with REST guidelines [31].

## 3. Results

### 3.1. Selection of Cohort

A total of 7883 patients with lung cancer were identified over the study period (Figure 1), of whom 225 were excluded as they had tracheal cancer (not shown in Figure 1). Separately, SEER registry matched 5540 of the 7883 UWM patients with lung cancer, of whom 1340 did not have a first primary tumor located in lungs and/or the histology code did not meet inclusion criteria and were excluded. Following linkage of the patients identified from the UW EDW (*n* = 7658) and those from SEER (*n* = 4200), a set of 4115 patients were identified common to both. We excluded patients who did not meet the ambulatory care definition (*n* = 3108), and those who had not received chest CT imaging at UWM (*n* = 243). Additional patients were excluded after review of missing or discrepant pathology data (*n* = 33) and those who lacked any ICD codes that could be used to calculate comorbidity (*n* = 20). The final cohort consisted of 711 patients.

### 3.2. Description of the Cohort

The mean age of the cohort was 67.3 years, 50% were female, the majority were non-Hispanic white (69.2%), with smaller numbers of Asian or Pacific Islander (11.3%) and non-Hispanic black (8.2%) (Table 1). At time of diagnosis, most patients were on Medicare (61.5%), and 14.9% living in a census tract where 20% or more inhabitants lived below the poverty line. Mean comorbidity score was 17.4, and 17.2% of patients had no history of smoking.

Of the included patients, 556 (78.2%) had NSCLC, 63 (8.9%) SCLC, 44 (6.2%) were categorized as other histology types, and 48 (6.8%) were unknown (Appendix B). Stage distribution was as follows: stage I 193 (27.1%), stage II 45 (6.3%), stage III 109 (15.3%), and stage IV 276 (38.8%) (8 (1.1%) individuals were stage 0, and 80 (11.3%) stage unknown). Individuals with late-stage (stages III or IV) cancer were significantly more likely to be male and have higher comorbidity scores than those with early stage (stages I or II) (Appendix C).

A total of 38 patients (5.3% of the cohort) met our definition for screen detected lung cancer, of whom 28 (75.7%) had NSCLC and 6 (16.2%) had SCLC. Their stage distribution was 18 (48.6%) stage I, 4 (10.8%) stage II, 6 (16.2%) stage III, 6 (16.2%) stage IV, and 3 (8.1%) were unknown. The vast majority of patients (36, 94.7%) whose lung cancer was identified by screening had recorded symptoms or signs associated with lung cancer documented prior to their lung cancer diagnosis.

### 3.3. Symptoms and Signs Prior to Diagnosis

The most common symptoms/signs prior to diagnosis were cough (573, 80.6%), shortness of breath (515, 72.4%), and fatigue (476, 67%) (Appendix D). Several symptoms/signs were significantly more frequent in individuals with early stage compared to late-stage cancer, namely cough (87.8% vs. 76.6%, *p* = 0.0008), shortness of breath (77.3% vs. 69.1%, *p* = 0.033), chest crackles or wheeze (62.2% vs. 50.9%, *p* = 0.008), and bone pain (47.9% vs. 34.0%, *p* = 0.0008). Lymphadenopathy was the only symptom/sign significantly more frequent in late stage than early stage (27.3% vs. 11.8%, *p* = 0.0000). Lymphadenopathy was the only clinical feature that was significantly more frequent in patients with SCLC compared to those with NSCLC (21 (33.3%) vs. 110 (20.1%), *p* = 0.024) (Appendix E).

### 3.4. Impact of Definition of Initial Symptomatic Presentation on Time to Diagnosis

The remainder of this analysis is limited to those patients (n = 647) who had one or more symptoms/signs (as defined above) and who were not diagnosed by LDCT screening. As the number of symptoms/signs used to define symptomatic presentation (Timepoint A) increased, the number of patients who fulfilled this criterion decreased, from 647 (with ≥1 symptom/sign, to 570 (≥2), 396 (≥3), 233 (≥4) to 122 (≥5) (Table 2, Figure 2). In addition, as the number of symptoms/signs used to define symptomatic presentation (Timepoint A) increased, the median number of days to diagnosis (i.e., interval from Timepoint A to D) decreased from 570 days (IQR 273, 690) for ≥ 1 symptom/sign to 265 days (IQR 148, 445) for ≥ 5 symptoms/signs.

### 3.5. Duration of Illness and Length of Key Time Intervals Prior to Diagnosis

Among the individuals (n = 647) who had one or more symptoms recorded prior to diagnosis, (Table 3 and Figure 3), the median time interval from initial clinical presentation to chest CT or chest X-ray imaging (interval from Timepoint A to B) was 291 days (IQR 144, 552), and from initial clinical presentation to specialist visit (A to C) was 236 days (IQR 118, 467), suggesting that some patients attended specialists prior to obtaining chest CT or chest X-ray imaging. The median duration between chest CT or chest X-ray imaging and diagnosis (Timepoints B to D) was 43 days (IQR 11, 240) and from specialist visit to diagnosis (C to D) 72 days (IQR 13, 456). Finally, the time interval from diagnosis to treatment initiation (D to E) was 12 days (IQR 0, 36).

Visualization of time intervals by stage of lung cancer (Figure 4) indicates longer median time interval A to D for early than late-stage cancer (639 vs. 540 days), including markedly longer intervals B to D (100 vs. 23 days) and C to D (244 vs. 36).

## 4. Discussion

### 4.1. Summary

As a first step in defining metrics that could be used to develop a CQM that would measure the timeliness of cancer diagnosis, we describe key time intervals from initial presentation to diagnosis of individuals with lung cancer. Our findings support the need for additional research and quality improvement efforts to improve early detection; the vast majority (94.5%) of patients were diagnosed following clinical presentation rather than by LDCT screening (5.5%), and the majority (54%) were diagnosed at a late stage (stages III or IV). Patients’ medical records showed evidence that one or more symptoms/signs associated with lung cancer were present a median of 570 days prior to diagnosis. This time interval was shorter when the presence of multiple symptoms was used to define initial presentation. The key time points of initial chest imaging and visits with specialists were overlapping rather than sequential as expected from previous literature. This implies that a CQM that uses imaging or specialist consultation as discrete timepoints will need to consider the complex nature of US healthcare, where the ‘gatekeeper’ role of primary care is often not well established and access to and/or co-management with specialists is not uncommon. While we observed longer time intervals (e.g., symptomatic presentation to diagnosis) for patients with early stage vs. late-stage cancer, intervals were overlapping and we could not identify definitive evidence of longer pre-diagnosis phases in individuals with later stage cancer. We might be able to determine the value of a CQM in this area if we were to do an intervention to alter these patterns and that intervention was successful. However, until there is clear evidence for an association between pre-diagnosis phase, stage of cancer and lung cancer outcomes, implementing a CQM with the sole intention of promoting a shift to earlier stage at diagnosis may be premature.

### 4.2. Comparison to Current Literature

The duration from symptom onset to diagnosis we identified is longer than most previous studies. The median interval reported by a study of Medicare claims data was 187 days (IQR 36-308), although this was limited to claims data and a period of 12 months before diagnosis (compared to our look-back period of 2 years) [9]. Several European studies using data from primary care describe time intervals from first documentation of coded symptoms to diagnosis of up to six months [8,32,33]. The intervals we identified exceed World Health Organization targets (90 days) and Canadian guidelines (4 weeks) from symptom onset to treatment and diagnosis of lung cancer, respectively [12,13,14]. In part, this pattern could derive from our setting: the UW is a major tertiary care center and transitions within the setting may reflect this structure.

The lack of association that we observed between time to diagnosis and stage is echoed in a recent study of 10,824 patients with NSCLC which found an inverse relationship between time to diagnosis and overall survival, even after adjusting for multiple potential confounders [34].

NLP identified a richer set of symptoms/signs from free text clinical fields compared to coded data [17]. This may explain why we identified earlier documentation of symptoms/signs potentially associated lung cancer than previous studies that did not use NLP methods. Interviews with patients who have recalled their early clinical presentations of lung cancer have highlighted a period of months or years prior to diagnosis, where individuals describe bodily changes, which may initially be dismissed or not attributed to cancer by patients or health care providers due to lack of awareness or fear of illness [35,36,37,38,39]. This important finding suggests unrealized value in considering CQMs here.

The time intervals we identified from initial chest imaging and/or consultation with specialists to diagnosis (43 and 71 days, respectively) are longer than those noted in some previous studies from the US. A recent study that used SEER-Medicare data for patients with NSCLC identified a median of 20 days between radiographic suspicion and diagnosis [34]. Another study described median intervals from abnormal chest imaging to treatment of 36.5 days, and specialist consultation to treatment of 27 days [40]. However, a small study of 129 Veterans noted a far longer median time from first chest imaging suspicious of cancer to treatment of approximately 3 months [41]. The short time interval we identified from diagnosis to treatment initiation (median of 7 days) is at the lower end of a range of previous reports of this time interval including a range of 6–45 days [42] and 22 days [43], and from specialist appointment to surgical intervention of 59 days [44]. The short duration we observed may be skewed by patients diagnosed at the time of surgery, rather than a measure of health system performance overall.

### 4.3. Strengths and Limitations

This is the first study in the US which has defined key diagnostic time intervals using EHR data and applied NLP to extract symptoms/signs that could be related to lung cancer prior to diagnosis. The cohort is representative of individuals who receive care in ambulatory settings in Washington State. We used a broad definition of ambulatory care, which included primary care and emergency medicine, reflecting US healthcare where some patients lack primary care providers. Our cohort is similar in terms of age, stage at diagnosis, and cancer type to studies from primary care settings in other countries [45]. In addition, the rates of lung cancer detected with LDCT screening we observed are consistent with contemporaneous data of screening rates of 3.9% among eligible adults [46]. Using NLP to extract details of symptoms and clinical features provided more detailed descriptions of clinical presentations than would have been possible using either coded or claims data alone as used in previous research [17]. Linkage to SEER cancer registry that has been used in multiple previous studies and provided diagnosis dates, staging, as well as key variables at time of diagnosis [9,47,48].

We acknowledge several limitations. First, the definition of our cohort in terms of ambulatory care relationship and requirement for chest CT may have biased selection of individuals with lung cancer, although the characteristics of our cohort are similar to those from previous studies. We are also aware that imaging tests in addition to chest CT are used in the diagnostic and staging workup of individuals with suspected lung cancer. While the demographics of our cohort is reflective of the population in Washington state, our cohort has fewer patients who identify as African American or those from rural areas than other regions in the US, therefore our findings may not be representative nationally or among patients not attending academic health science centers. This is important as there is evidence of disparities in cancer diagnosis among rural and other underserved communities in the US [49]. In addition, our study is based on a cohort from a single site, which further limits its generalizability. Second, while we attempted to extract all available EHR records on the cohort, patients may have entered or exited the UWM system during the study period due to changing health insurance and/or residence, and thus we may not have accessed the entirety of their health records [50]. Missing data due to care received at non-UWM sites could have altered the time intervals. However, we took special care to exclude patients as non-informative who may not have received their pre-diagnosis care at UWM. Furthermore, NLP extraction is limited by performance of the annotation tool used and what the provider documents, which could vary widely from provider to provider. Third, our definition of onset of clinical presentation based on the documentation of a certain number of symptoms/signs associated with lung cancer may be too non-specific and merely reflect other concomitant illness, however these clinical features are consistent with multiple other studies that have examined clinical presentation of lung cancer in ambulatory care [6,17,51].

### 4.4. Study Implications

Most ambulatory care providers receive little feedback on their diagnostic performance for serious (but rare) conditions such as cancer; a CQM could help to inform practitioner and clinic-wide efforts to improve practice. Our study demonstrates that key clinical features are recorded by clinicians in the EHR for a considerable period prior to diagnosis among patients who are later diagnosed with lung cancer. Tools could be implemented in EHR data to flag clinical signs that raise the probability of lung cancer, potentially using sophisticated models, although this would need to be balanced with the risks of unnecessary investigations and referrals. The time intervals we describe, if validated by other research, could inform upper limits of intervals for several steps in the pre-diagnosis period as part of a CQM. Interventional studies will be needed to determine the impacts of such measures on time to diagnosis, stage, overall survival as well as unintended negative impacts on healthcare utilization. However, we found few differences in symptoms/signs with stage at diagnosis; other studies suggest the symptom burden is indeed higher in patients with more advanced stage but that there may not be an association between longer diagnostic interval and later stage disease [52,53].

## 5. Conclusions

It is surprising that the US has no widely accepted CQM for the diagnosis of lung cancer, despite the burden this disease causes to patients and the healthcare system. Our findings suggest that many patients have a long symptomatic period, prior to diagnostic testing and specialty visits, suggesting potential for interventions to improve timeliness of diagnosis, and potential for improving outcomes. Efforts are needed to develop and test interventions that can be applied in ambulatory care settings to improve the detection of individuals with lung cancer.

## Figures and Tables

**Figure 1 cancers-14-05756-f001:**
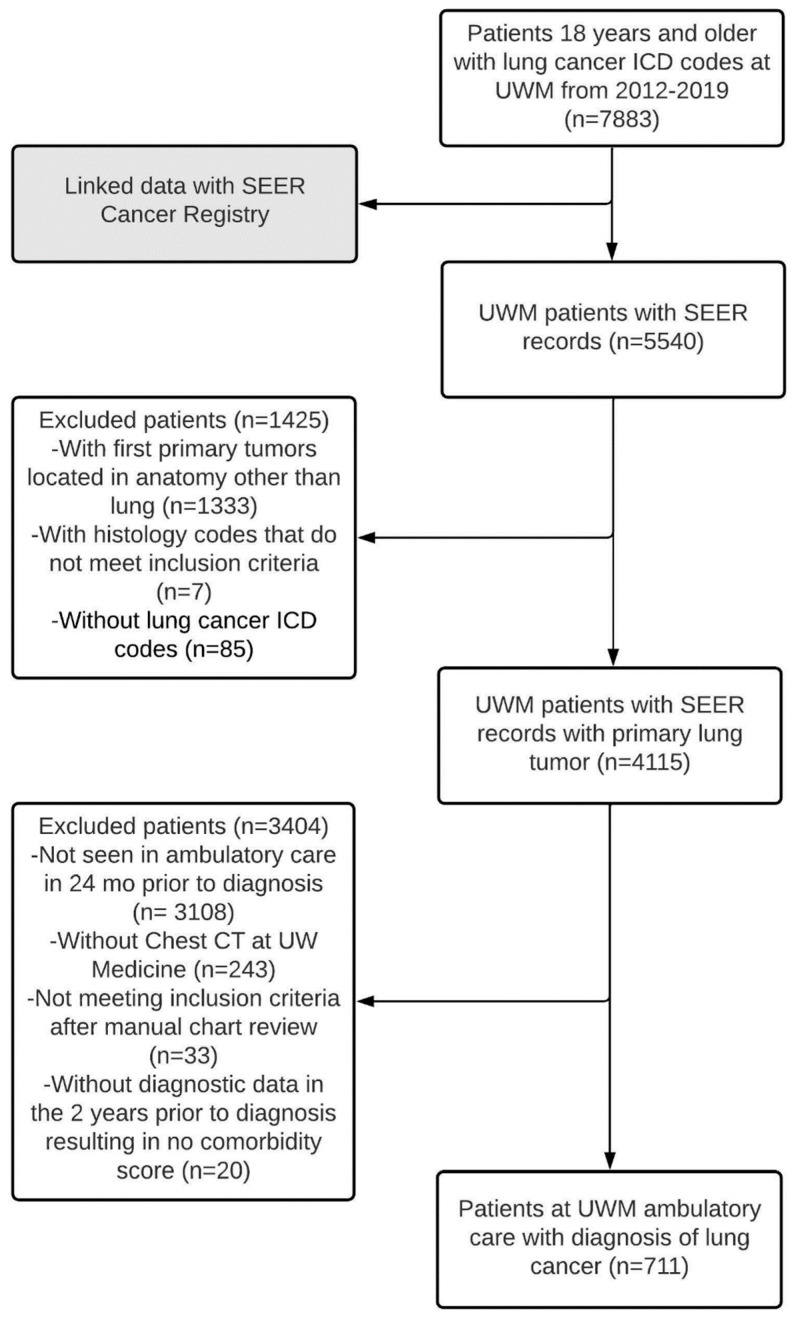
Flow chart of selection of patients with lung cancer.

**Figure 2 cancers-14-05756-f002:**
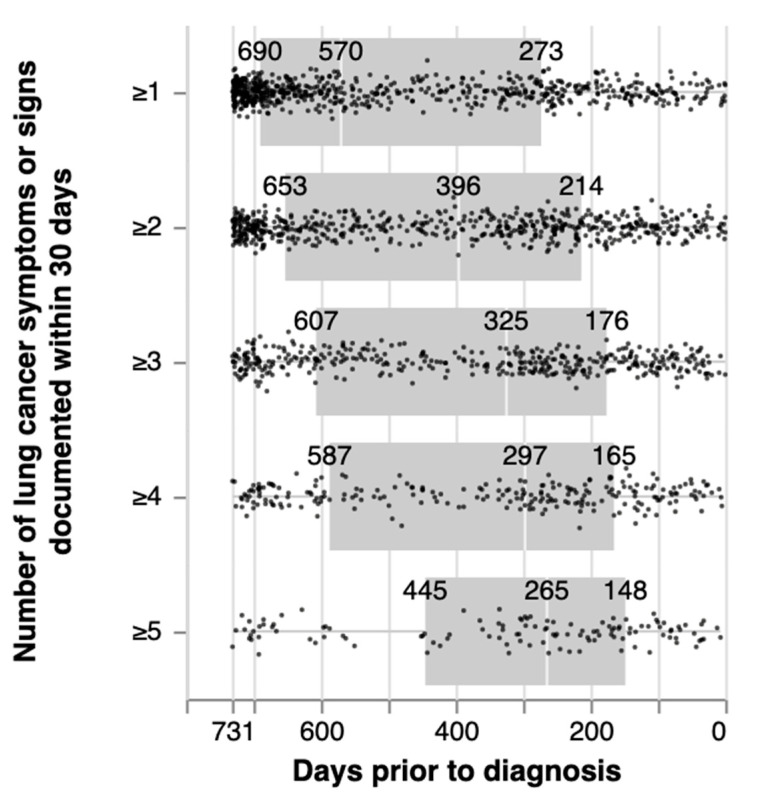
Length of time between first recorded symptom/sign potentially associated with lung cancer within two years (24 months) of diagnosis and date of diagnosis. Note: 30-day window means x-number of symptoms recorded in the EHR within 30 days of one another, not necessarily the same visit.

**Figure 3 cancers-14-05756-f003:**
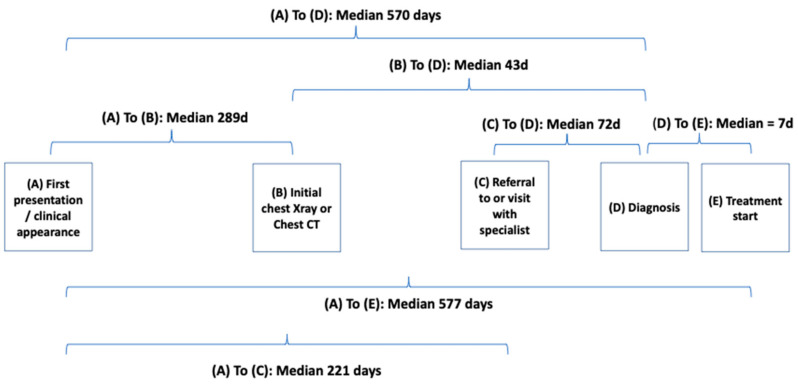
Duration of time intervals from first symptomatic presentation to diagnosis and initiation of treatment.

**Figure 4 cancers-14-05756-f004:**
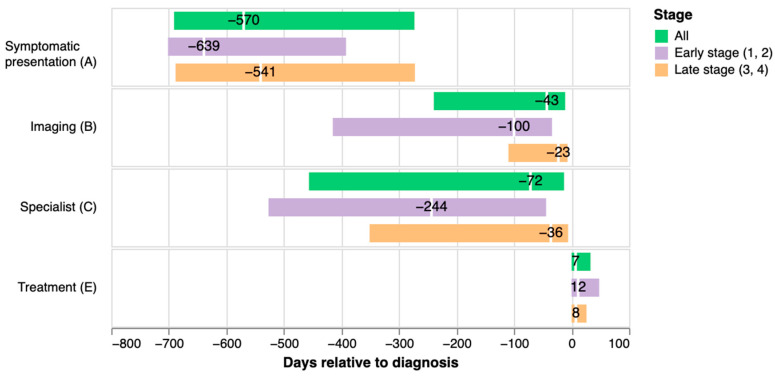
Median number of days prior to diagnosis for initial symptomatic presentation, chest CT or chest X-ray imaging, cancer specialist visit, and after diagnosis for initiation of treatment by stage of lung cancer. Note: Imaging includes chest X-ray or chest CT.

**Table 1 cancers-14-05756-t001:** Characteristics of patients with lung cancer by stage at diagnosis.

Patient Characteristics	All (n = 711) n (%) *	Stage 1 (n = 193) n (%)	Stage 2 (n = 45) n (%)	Stage 3 (n = 109) n (%)	Stage 4 (n = 276) n (%)	Stage Not Known (n = 80) n (%)
Age						
18–49	36 (5.1)	7 (3.6)	2 (4.4)	4 (3.7)	19 (6.9)	4 (5.0)
50–59	129 (18.1)	30 (15.5)	7 (15.6)	23 (21.1)	49 (17.8)	17 (21.2)
60–69	261 (36.7)	74 (38.3)	16 (35.6)	43 (39.4)	98 (35.5)	26 (32.5)
70–79	185 (26.0)	51 (26.4)	15 (33.3)	28 (25.7)	72 (26.1)	19 (23.8)
80+	100 (14.1)	31 (16.1)	5 (11.1)	11 (10.1)	38 (13.8)	14 (17.5)
Sex						
Male	355 (49.9)	73 (37.8)	29 (64.4)	61 (56.0)	145 (52.5)	43 (53.8)
Race/Ethnicity						
Asian or Pacific Islander	80 (11.3)	17 (8.8)	8 (17.8)	11 (10.1)	33 (12.0)	11 (13.8)
Hispanic or Latino	23 (3.2)	6 (3.1)	3 (6.7)	5 (4.6)	7 (2.5)	2 (2.5)
Non-Hispanic Black	58 (8.2)	21 (10.9)	3 (6.7)	8 (7.3)	23 (8.3)	3 (3.8)
Non-Hispanic White	492 (69.2)	144 (74.6)	27 (60.0)	80 (73.4)	179 (64.9)	56 (70.0)
Other	58 (8.2)	5 (2.6)	4 (8.9)	5 (4.6)	34 (12.3)	8 (10.0)
Smoking status						
Ever smoker	531 (74.7)	152 (78.8)	39 (86.7)	94 (86.2)	184 (66.7)	56 (70.0)
Never smoker	122 (17.2)	38 (19.7)	4 (8.9)	8 (7.3)	63 (22.8)	8 (10.0)
Unknown	58 (8.2)	3 (1.6)	2 (4.4)	7 (6.4)	29 (10.5)	16 (20.0)
Insurance						
Medicaid	117 (16.5)	25 (13.0)	3 (6.7)	25 (22.9)	49 (17.8)	15 (18.8)
Medicare	437 (61.5)	133 (68.9)	32 (71.1)	58 (53.2)	164 (59.4)	45 (56.2)
Military	13 (1.8)	5 (2.6)	0 (0.0)	3 (2.8)	4 (1.4)	1 (1.2)
Not Insured	7 (1.0)	1 (0.5)	0 (0.0)	3 (2.8)	3 (1.1)	0 (0.0)
Private	130 (18.3)	28 (14.5)	10 (22.2)	19 (17.4)	54 (19.6)	17 (21.2)
Unknown	7 (1.0)	1 (0.5)	0 (0.0)	1 (0.9)	2 (0.7)	2 (2.5)
Census Tract Poverty Indicator						
0–10% poverty	383 (53.9)	108 (56.0)	21 (46.7)	56 (51.4)	151 (54.7)	41 (51.2)
10–20% poverty	222 (31.2)	53 (27.5)	15 (33.3)	31 (28.4)	92 (33.3)	29 (36.2)
≥20% poverty	106 (14.9)	32 (16.6)	9 (20.0)	22 (20.2)	33 (12.0)	10 (12.5)
Comorbidity: Elixhauser van Walraven Weighted Score mean (SD)	17.36 (11.8)	13.53 (9.8)	15.76 (11.8)	16.27 (12.1)	21.19 (11.9)	16.16 (12.0)

* Individuals with Stage 0 (n = 8) excluded from this table.

**Table 2 cancers-14-05756-t002:** Length of time (days) between first recorded lung cancer clinical features within two years (24 months) of diagnosis and date of diagnosis.

Number of Symptoms/Signs Present within 30-Day Window *	Number of Patients	Mean (SD)	Range (Shortest, Longest Interval)	Median (IQR)
≥1	647	481 (228)	0, 731	570 (273–691)
≥2	570	412 (233)	0, 731	396 (213–653)
≥3	396	377 (230)	0, 731	322 (176–607)
≥4	233	355 (225)	5, 731	297 (165–587)
≥5	122	314 (217)	7, 731	264 (148–445)

Note: Excludes cohort members with lung cancer detected by LDCT and those without any symptoms/signs. * 30-day window means x-number of symptoms recorded in the EHR within 30 days of one another, not necessarily the same visit.

**Table 3 cancers-14-05756-t003:** Duration of time intervals from first symptomatic presentation, chest imaging, specialist visit, diagnosis, and initiation of treatment for lung cancer *.

	All		Cancer Type				Stage			
			NSCLC		SCLC		Stages 1,2		Stages 3,4	
	*n*	Median (IQR)	*n*	Median (IQR)	*	Median (IQR)	*n*	Median (IQR)	*n*	Median (IQR)
Interval										
A to D (Presentation to diagnosis)	647	570 (273–691)	504	584 (305–691)	57	605 (314–709)	211	639 (392–702)	356	540 (272–688)
A to B (Presentation to chest imaging ^†^)	635	291 (144–552)	497	313 (149–559)	57	307 (183–627)	209	286 (134–536)	348	324 (176–586)
A to C (Presentation to specialist visit)	640	236 (118–467)	499	250 (123–491)	57	203 (93–488)	210	216 (114–480)	352	261 (129–522)
B to D (Chest imaging to diagnosis)	635	43 (11–240)	497	44 (14–255)	57	43 (10–150)	209	100 (34–415)	348	23 (7–110)
C to D (Specialist visit to diagnosis)	640	72 (13–456)	499	87 (15–468)	57	84 (7–429)	210	244 (44–527)	352	36 (7–351)
D to E (Diagnosis to treatment initiation) **	525	12 (0–36)	425	13 (0–40)	51	3 (0–13)	188	19 (0–49)	282	9 (0–28)

* Table presents data only on the cohort who had one or more symptoms, and who were not diagnosed by LDCT screening. ^†^ Chest Imaging includes chest X-ray, Chest computerized tomography (CT) scan, or both. ** Date of treatment initiation missing for 109 who had no treatment given, 7 had active surveillance (watchful waiting), and 6 had treatment indicated but a start date could not be identified.

## Data Availability

Fully anonymized data may be available on reasonable request to the corresponding author, once appropriate data sharing and ethics approvals have been obtained.

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
