# Peer review of "How Timely Is Diagnosis of Lung Cancer? Cohort Study of Individuals with Lung Cancer Presenting in Ambulatory Care in the United States"

_cancers, 2022, doi:10.3390/cancers14235756_

Round 1
Reviewer 1 Report
I would like to congratulate the authors for such an interesting and complete work. It is a paramount aspect to an optimal lung cancer treatment to reduce the times between the first lung cancer symptoms onset to its treatment, and this paper gives several ideas which could improve it.
After its reading, I have detected one important bias and some minor details.
The important bias is the definition of “diagnosis”; the authors have defined it just as having a CT scan done. Since the study covers a great amount of time (from 2012 to 2019), maybe, it should be differentiated if patients have had PET-CT scan done, bronchoscopy, biopsy, etc… Maybe, for example, PET-CT wasn’t as affordable in 2012 as it is now, and it is likely that not as many patients back then had it done as they do nowadays, so it could be a potential bias, since it will probably increase the diagnostic time. It would be interesting that authors would differentiate between with pre-diagnostic tests have patients undergone before reaching their treatment.
On the other hand, authors cite several appendixes, but they are not between the sent material, and neither is the link attached
The other minor observations would be:
- TITLE: I would recommend a shorter tittle; try to sum it.
- LINE 225: “Several symptoms/signs 225 were significantly more frequent in individuals with early-stage compared to late-stage”: This seems to be against the stablished bibliography; how would you explain it? You should compare it with other author’s experience.
- FIGURE 2: It would be more easily understandable if in x axis, 0 would be at the intersection with y, so the number of days to diagnosis would increase to the right.
- REFERENCE: Number 13 has not DOI or PMID. Besides, in works with more than 10 authors, it would be advisable not to write them all, but to write “et al.” after the 10th.
Author Response
REVIEWER 1
Reviewer comment
I would like to congratulate the authors for such an interesting and complete work. It is a paramount aspect to an optimal lung cancer treatment to reduce the times between the first lung cancer symptoms onset to its treatment, and this paper gives several ideas which could improve it.
Author response
Thank you for this.
Reviewer comment
After its reading, I have detected one important bias and some minor details.
The important bias is the definition of “diagnosis”; the authors have defined it just as having a CT scan done. Since the study covers a great amount of time (from 2012 to 2019), maybe, it should be differentiated if patients have had PET-CT scan done, bronchoscopy, biopsy, etc.… Maybe, for example, PET-CT wasn’t as affordable in 2012 as it is now, and it is likely that not as many patients back then had it done as they do nowadays, so it could be a potential bias, since it will probably increase the diagnostic time. It would be interesting that authors would differentiate between with pre-diagnostic tests have patients undergone before reaching their treatment.
Author response
Diagnosis of lung cancer was based on the entry within the NCI-funded SEER cancer registry, which is used in the counties surrounding Seattle. The SEER registry was used to confirm the diagnosis of lung cancer specifically (excluding certain cancers e.g. mesothelioma etc., as noted in the Methods page 7) as well as the date of diagnosis. We further cross checked the date of diagnosis between the SEER cancer registry and the entry in the UW Medicine electronic medical record (EHR) for each patient. In cases where there was a discrepancy in dates between the date in SEER and the date in the EHR, we manually checked the patient’s notes to determine date of diagnosis – Appendix A provides the details of how Date of Diagnosis (as well as other timepoints) were defined.
We did use the presence of a CT Chest in the patient’s EHR to verify that the patient was a patient whose work up and imaging for lung cancer had occurred within UW Medicine, based on the fact that it is not possible over the period of time of this study for a diagnosis of lung cancer to have been made without a CT chest. The reviewer is correct that additional tests like PET are now used as well as CT chest, but do not replace use of CT chest for initial diagnosis purposes. This detail is also provided in Appendix A
Reviewer comment
On the other hand, authors cite several appendixes, but they are not between the sent material, and neither is the link attached
Author response
We have the appendices available on the uploaded journal system, and should be available for the Reviewers and other readers.
Reviewer comment
The other minor observations would be:
1.TITLE: I would recommend a shorter tittle; try to sum it.
- LINE 225: “Several symptoms/signs 225 were significantly more frequent in individuals with early-stage compared to late-stage”: This seems to be against the stablished bibliography; how would you explain it? You should compare it with other author’s experience.
- FIGURE 2: It would be more easily understandable if in x axis, 0 would be at the intersection with y, so the number of days to diagnosis would increase to the right.
- REFERENCE: Number 13 has not DOI or PMID. Besides, in works with more than 10 authors, it would be advisable not to write them all, but to write “et al.” after the 10th.
Author response
1.TITLE: We are open to changing the title, suggested title now included in the revised manuscript is as follows: How timely is diagnosis of lung cancer? Cohort study of individuals with lung cancer presenting in ambulatory care in the United States
- LINE 225. The association between number and type of symptoms and stage of cancer is complicated. There is some evidence from other studies that individuals with early-stage cancer have more symptoms, hence they are diagnosed more expeditiously than those with later stage cancer. Furthermore our study is the first that has used NLP to drill down into the detailed symptom information that is contained within the free text of clinician’s notes, and hence our findings may run contrary to other studies due to this highly detailed clinical phenotyping.
- FIGURE 2: We would prefer to leave this figure as it is. It provides a descriptive summary of the distribution of length of time between symptoms and date of diagnosis
- REFERENCE: We have added the PMID (there is no DOI for this publication) to the revised manuscript.

Reviewer 2 Report
The authors present the results of a descriptive study assessing the local time intervals in the diagnostic and treatment phases of lung cancer patients that were seen in ambulatory care at UWM. They extracted symptoms and signs from the EMR to determine diagnostic intervals from first symptom presentation to diagnosis. This study highlights the challenges in research assessing timeliness of care measures for patients with lung cancer. The authors suggest that the results of this study could be used to guide clinical quality measures. However, from a quality improvement perspective, the scope of the patient population considered is too limited to guide CQM, and rather the paper highlights the reasons for which developing quality measures around timeliness of care has been so challenging. The paper would benefit from revisions to better clarify the descriptive nature of the study and highlight the challenges around developing timeliness of care measures, rather than suggesting that the results of this study could guide the development of quality measures.
Abstract - “Symptoms/signs associated with lung cancer can be identified over a year prior to diagnosis using NLP, highlighting the need for CQMs to improve timeliness of diagnosis”. The statement that the fact that these symptoms are present for over a year prior to diagnosis using NPL highlights “the need for CQM to improve timelines of diagnosis” is a leap and should be revised. As will come up frequently in the review of the main body of the paper, many lung cancer symptoms are non-specific. How would a CQM for timeliness of care address the fact that these symptoms aren’t specific for lung cancer? There is certainly value in early detection, but this inference makes too strong of an association between non-specific symptoms at presentation and the need for timely care.
Introduction
Page 2, line 83. “while guidelines in Canada recommend time from presentation to diagnosis should be a maximum of four weeks [13-16]” Please note that one of the cited articles is Danish, not Canadian.
Materials and Methods
2.1 A limitation to the study is that it is a cohort study based out of a single health system in one city and state. It is difficult to generalize across an entire country. The data from this study alone cannot be used to generate a CQM that applies to the whole country. While this comes up later in the limitations section, it is critical to highlight that the results from this localized study cannot be used to generate a CQM that would apply to the entire US population as the authors seem to suggest. Rather, the authors should highlight that this is a descriptive study of local timeliness of care, which may be able to suggest opportunities for improvement at a local level, but cannot be generalized to opportunities for improvement across the country.
2.5 Can the authors explain why they selected a specialist visit with obstetrics and gynecology as meeting criteria for an ambulatory care relationship? It would seem that this type of specialist visit would be irrelevant to the timely diagnosis of lung cancer. Similarly, while I’m not sure of the specific scope of specialists in “women’s health” this could be further elaborated upon.
Line 154 – Appendix A does not accompany the available materials for review. In fact, none of the Appendices are included with the review material.
Results
Why were there so many patients identified as having lung cancer by UWM that were not able to be matched in SEER? – 2343 patients were not identified in SEER, and an additional 1425 excluded as not having lung cancer – can the authors please explain why such a substantial subset of patients were excluded? What is the implication on developing timeliness of care targets if the selected population if just over half of the total patient population at UWM identified as having lung cancer?
Meanwhile, the final cohort of patients used in the study consisted of 711, which is less than 10% of the group of patients identified by UWM as having lung cancer. This is a major limitation to the study – CQM related to lung cancer care need to be generalizable, equitable, and applicable to the many varied presentations of patients with lung cancer. A cohort of patients of 10% of the target population is so limited as to not be practical. This needs to be expanded upon in the discussion and limitations.
Furthermore, while the ambulatory care definition is certainly important, the fact that 3108 patients were excluded as they were not seen in ambulatory care highlights a major gap in care – we need to address the fact that so many of these patients are diagnosed in acute care and other pathways. CQMs that do not consider the fact that most patients present with advanced stage disease and urgent presentation will not address the quality concerns related to lung cancer care.
It also concerns me that patients receiving Chest CT outside of UW Medicine are excluded as this does not factor in the real-life consideration that patients receive tests at multiple different institutions, and CQMs should take into consideration the many different pathways by which patients receive care – either by standardizing pathways or addressing barriers to why pathways are varied.
As such, due to the significant exclusions in this study, the authors should present the data as descriptive of a specific pathway of care, rather than try to make inferences about how the analysis of timeliness of care from this very limited population (10% of patients coded as having LC at UWM) can be used to generate CQMs that apply to the whole US population.
Table 1 - Please explain why such a large proportion of patients are classified as ”stage not known”. How might this affect the data?
Please elaborate on the implications of lumping screen detected cancers and non-screen detected cancers in this study and reconsider how these patients are analysis/ included in the study. By definition, patients with screen detected cancers should be asymptomatic, or at least not have symptoms suggestive of lung cancer. Does including these patients affect the timeliness of care measures from symptom onset to various diagnostic milestones? Should screen detected lung cancers be considered a separate group when developing CQMs for timeliness of care? It would seem that these patients should be excluded from the analysis of the time points from symptoms and signs prior to presentation.
- It is noted that asymptomatic and screen-detected patients were excluded from the analysis from section 3.4 moving forward. However, it raises the question as to whether they should have been included in the study at all, even in terms of assessment of symptoms and signs prior to diagnosis, as this should be a different patient population than non-screen detected cancers.
- Having said that, while screen detected patients are excluded for the rest of the analysis, it would be more important to include these patients in the analysis of time from abnormal imaging to diagnosis and time to treatment. Once a patient has an abnormal CT scan, the pathways for screen detected and non-screen detected patients should proceed similarly. It’s the interval from symptoms to diagnosis that would be most impacted by screen-detected vs. non-screen detected differences.
2.2 – Regarding symptoms and signs prior to diagnosis:
- What percentage of patients with the listed symptoms in section 2.5 did not have lung cancer? What is the positive predictive value of the symptoms listed in this study?
- Please explain why individuals with early-stage disease were more likely to complain of bone pain – this does not make sense clinically as early stage patients wouldn’t have bony disease from lung cancer. As such, is it relevant as a sign/symptom if it does not correlate with lung cancer?
Line 225 – Appendix D does not accompany the paper review materials
I don’t think Table 2 adds anything beyond what Figure 2 shows.
This paper is also limited in its lack of evaluation of outcomes related to timely care. What is the impact of delays in care on survival outcomes for various stages of disease? How do we know that faster care is better if the data shows the effects of “sicker quicker” care i.e. that the most symptomatic patients receive the fastest care?
Discussion
Rather than frame this study as informing metrics that could be used to develop a CQM that would measure the timeliness of cancer diagnosis, I think it would better to present this as a paper that describes a single centre experience in timeliness of care for ambulatory lung cancer patients. It is not a fulsome enough evaluation to be able to inform a CQM around timeliness of care.
This study does not add a lot to the literature – it is known that diagnostic intervals for lung cancer are too long, that the sicker-quicker effect exists whereby patients with advanced, urgent presentations receive faster care, and that many patients present with late-stage disease. What this study shows is the utility in local quality assurance processes of measuring timeliness of care, but it cannot inform system wide improvement strategies given the limited scope.
Line 300 “However, until there is clear evidence for an association between pre-diagnosis phase and stage of cancer, implementing a CQM with the sole intention of promoting a shift to earlier stage at diagnosis may be premature”. I suggest adding that “until there is clear evidence for an association between pre-diagnosis phase, stage of cancer, and lung cancer outcomes”. We know that earlier stage disease is associated with better outcomes, but we don’t know the impact of timeliness of care on survival – i.e. the extent to which delays in care impact survival, for which stages of disease, and what constitutes a significant delay likely to impact survival.
Line 310 – again the authors reference a Danish study when referring to WHO targets and Canadian guidelines
Line 325 – “This important finding suggests unrealized value in considering CQMs here”. I find it challenging to understand how non-specific symptoms such as cough and fatigue can be used to guide a CQM for lung cancer. Even more so, the fact that symptoms like bone pain were more common in early stage disease (without bony metastatic disease by definition) seem to suggest that some symptoms may be unreliable in terms of facilitating a diagnosis. Is the time intervals from non-specific symptoms to diagnosis really the best quality measure to guide lung cancer care?
Line 226 – “The short duration we observed may be skewed by patients diagnosed at the time of surgery, rather than a measure of health system performance overall.” This is a methodological flaw in the analysis, and one of the reasons these types of analyses are so difficult. In the case of patients going to surgery, one strategy is to consider the date of decision to operate as a date of diagnosis, to better understand the time intervals. It would also be important to consider timeliness of care measures by lung cancer stage to help sort out to what extent the early stage group diagnosed at surgery impacted this measure vs. the sicker quicker effect whereby patients with more advance and more symptomatic disease received faster care.
From a QI perspective, the better CQM measures would be time from abnormal imaging to diagnosis (either rule in lung cancer or rule out) – not just time to diagnosis of lung cancer patients alone. We also need to consider the value of other quality metrics beyond timeliness of care (efficiency of resource utilization, % patients receiving guideline recommended (effective) care, equitability of care). I think the paper needs to include more information in the discussion about how a) timeliness of care measures are challenging for lung cancer care given the many challenges identified in this study, b) we should consider to what extent there are other measures of quality of lung cancer care that should be monitored.
Line 387 – “However, we found few differences in symptoms/signs with stage at diagnosis”. Is it not concerning that the study did not find that symptoms correlated with stage of disease – it makes one wonder whether the symptoms were associated with lung cancer at all.
Conclusion line 395 – “It is surprising that the US has no widely accepted CQM for the diagnosis of lung 395 cancer, despite” – Is it surprising? This study shows the challenges with regards to establishing timeliness of care metrics, particularly in terms of the challenges in obtaining data, determine time intervals, and sorting out the “sicker quicker” effect.
399 – “and potential for improving outcomes”. – I think it is important to note that this study does not address outcomes at all.
Author Response
REVIEWER 2
Reviewer comment
The authors present the results of a descriptive study assessing the local time intervals in the diagnostic and treatment phases of lung cancer patients that were seen in ambulatory care at UWM. They extracted symptoms and signs from the EMR to determine diagnostic intervals from first symptom presentation to diagnosis. This study highlights the challenges in research assessing timeliness of care measures for patients with lung cancer. The authors suggest that the results of this study could be used to guide clinical quality measures. However, from a quality improvement perspective, the scope of the patient population considered is too limited to guide CQM, and rather the paper highlights the reasons for which developing quality measures around timeliness of care has been so challenging. The paper would benefit from revisions to better clarify the descriptive nature of the study and highlight the challenges around developing timeliness of care measures, rather than suggesting that the results of this study could guide the development of quality measures.
Author response
We agree that it is important to highlight that this was a single site, and merely provides initial evidence to inform potential development of a CQM, rather than attempting to provide the only evidence base that is needed in measure development.
We have modified the Introduction, page 7 as follows:
The overall aim of this study was therefore to define key time points and describe intervals in the diagnostic pathway of patients with lung cancer based on a single site in the US, from initial symptomatic presentation in ambulatory care to diagnosis, using routine EHR data. We believe our findings provide valuable new information that not only describe the timeliness of the diagnostic process for lung cancer, but could also potentially be used to inform efforts to develop CQM for lung cancer in a range of health care settings
Reviewer comment
Abstract - “Symptoms/signs associated with lung cancer can be identified over a year prior to diagnosis using NLP, highlighting the need for CQMs to improve timeliness of diagnosis”. The statement that the fact that these symptoms are present for over a year prior to diagnosis using NPL highlights “the need for CQM to improve timelines of diagnosis” is a leap and should be revised. As will come up frequently in the review of the main body of the paper, many lung cancer symptoms are non-specific. How would a CQM for timeliness of care address the fact that these symptoms aren’t specific for lung cancer? There is certainly value in early detection, but this inference makes too strong of an association between non-specific symptoms at presentation and the need for timely care.
Author response
We respond to this issue of non-specific symptoms in multiple responses below and direct the reviewer to the following responses. Thank you for your comment. To clarify, we feel that drawing attention to the length of time that non-specific symptoms present prior to diagnosis illustrates how a CQM may be beneficial, and that in turn providers may further investigate non-specific symptoms.
Reviewer comment
Introduction
Page 2, line 83. “while guidelines in Canada recommend time from presentation to diagnosis should be a maximum of four weeks [13-16]” Please note that one of the cited articles is Danish, not Canadian.
Author response
Thank you for pointing out this error in citation, which we have corrected and now added in a further sentence describing the Danish experience with lung cancer indicators (reference 14).
Reviewer comment
Materials and Methods
2.1 A limitation to the study is that it is a cohort study based out of a single health system in one city and state. It is difficult to generalize across an entire country. The data from this study alone cannot be used to generate a CQM that applies to the whole country. While this comes up later in the limitations section, it is critical to highlight that the results from this localized study cannot be used to generate a CQM that would apply to the entire US population as the authors seem to suggest. Rather, the authors should highlight that this is a descriptive study of local timeliness of care, which may be able to suggest opportunities for improvement at a local level, but cannot be generalized to opportunities for improvement across the country.
2.5 Can the authors explain why they selected a specialist visit with obstetrics and gynecology as meeting criteria for an ambulatory care relationship? It would seem that this type of specialist visit would be irrelevant to the timely diagnosis of lung cancer. Similarly, while I’m not sure of the specific scope of specialists in “women’s health” this could be further elaborated upon.
Line 154 – Appendix A does not accompany the available materials for review. In fact, none of the Appendices are included with the review material.
Author response
2.1: We agree that our study is based at a single site in the US, and acknowledge in both the Introduction (edited in response to a comment above), and have added a further point on this in the Discussion section page 19.
2.5: Thank you for this question. It is common for OB/GYN or Women’s Health to be a source of primary care for some women in the USA. Most studies that define primary care in the US would include OB/Gyn as one part of the primary care system. We realize that this is different in other countries’ definitions of primary care.
Line 154- we have provided the Appendices for Reviewers and readers
Reviewer comment
Results
Why were there so many patients identified as having lung cancer by UWM that were not able to be matched in SEER? – 2343 patients were not identified in SEER, and an additional 1425 excluded as not having lung cancer – can the authors please explain why such a substantial subset of patients were excluded? What is the implication on developing timeliness of care targets if the selected population if just over half of the total patient population at UWM identified as having lung cancer?
Author response
We provide the detailed study flowchart of participants in Figure 1. Our goal in identifying a cohort of individuals who had an established relationship with UW Medicine ambulatory care clinics for the 2 years prior to diagnosis was to assemble a cohort who could potentially provide data on the clinical presentations, referrals, imaging and workup up to the point of diagnostic confirmation. We fully acknowledge that many patients with lung cancer are referred into the highly specialized services provided to patients with cancer at UW Medicine, which acts as the main tertiary referral center for the state of Washington and many of the surrounding states (covering 20% of the US landmass), However, the rationale for this selection and the detailed inclusion and exclusion criteria provide justification for this. We agree that further studies to replicate our findings in other ambulatory care populations would be helpful.
Reviewer comment
Meanwhile, the final cohort of patients used in the study consisted of 711, which is less than 10% of the group of patients identified by UWM as having lung cancer. This is a major limitation to the study – CQM related to lung cancer care need to be generalizable, equitable, and applicable to the many varied presentations of patients with lung cancer. A cohort of patients of 10% of the target population is so limited as to not be practical. This needs to be expanded upon in the discussion and limitations.
Author response
We would certainly agree that efforts to improve the timeliness of lung cancer should be generalizable and equitable, but we feel it was an entirely justifiable to focus this initial study on those patients who present to ambulatory care settings (including ER) who are then diagnosed with lung cancer. This is the most common route for diagnosis. The vast majority of the patients excluded in Figure 1 (n= 3108) were those who had been referred in to the UW Medicine cancer specialists from outside clinics. In the perfect world it would be interesting to have obtained all of their primary care records and data for the period 2 years before their diagnosis, but that is currently impossible with the disparate electronic health record systems used in the USA. We note the limitations in sample size and need for replication of these findings in other ambulatory care populations in the US.
Reviewer comment
Furthermore, while the ambulatory care definition is certainly important, the fact that 3108 patients were excluded as they were not seen in ambulatory care highlights a major gap in care – we need to address the fact that so many of these patients are diagnosed in acute care and other pathways. CQMs that do not consider the fact that most patients present with advanced stage disease and urgent presentation will not address the quality concerns related to lung cancer care.
Author response
To clarify, we aimed to recruit patients who had an established relationship with ambulatory care, including ER, within our health care system. We therefore will certainly have included patients who presented to the ER with severe or late symptoms as part of this cohort. As noted above, many patients are referred into UW Medicine for specialty services (e.g. cancer care) from the state and Pacific North West region.
Reviewer comment
It also concerns me that patients receiving Chest CT outside of UW Medicine are excluded as this does not factor in the real-life consideration that patients receive tests at multiple different institutions, and CQMs should take into consideration the many different pathways by which patients receive care – either by standardizing pathways or addressing barriers to why pathways are varied.
Author response
We attempted to be rigorous in our inclusion to ensure valid results as we do not have access to all records outside of UW Medicine, and therefore could not verify imaging information if needed. We agree that “CQMs should take into consideration the many different pathways by which patients receive care” and hope this point can be incorporated as we build on this work. As noted in the Discussion section, our work highlights the challenges of generating evidence to inform a CQM in the US particularly, in comparison to several other developed nations where this is standard practice (e.g. Denmark, UK), largely due to data availability.
Reviewer comment
As such, due to the significant exclusions in this study, the authors should present the data as descriptive of a specific pathway of care, rather than try to make inferences about how the analysis of timeliness of care from this very limited population (10% of patients coded as having LC at UWM) can be used to generate CQMs that apply to the whole US population.
Author response
There is no subjective judgment regarding the timeliness or lack of timeliness based on the data, and as such we feel we presented the data as descriptive. We did not seek to generate a CQM that would apply to the ‘whole US population’, but rather patients presenting in ambulatory care settings. This comment is addressed in the limitations sections.
Reviewer comment
Table 1 - Please explain why such a large proportion of patients are classified as ”stage not known”. How might this affect the data?
Author response
There were 80 patients whose stage was not known as noted in Table 1. The information on cancer staging is provided by the SEER cancer registry, and we were not able to investigate further reasons for this. From other work that has used SEER, unknown stage can occur among patients who have more palliative treatment/diagnosis care. We did not have sufficient sample size to conduct subgroup analyses to explore their potential similarities to the rest of the cohort in whom we did have stage.
Reviewer comment
Please elaborate on the implications of lumping screen detected cancers and non-screen detected cancers in this study and reconsider how these patients are analysis/ included in the study. By definition, patients with screen detected cancers should be asymptomatic, or at least not have symptoms suggestive of lung cancer. Does including these patients affect the timeliness of care measures from symptom onset to various diagnostic milestones? Should screen detected lung cancers be considered a separate group when developing CQMs for timeliness of care? It would seem that these patients should be excluded from the analysis of the time points from symptoms and signs prior to presentation.
Author response
We describe the characteristics of the cohort, regardless of method/route of diagnosis in table 1, this includes those diagnosed after symptomatic presentation and those diagnosed after screening. This seems completely reasonable. However, as noted on page 13, Section 3.4, the remainder of the analysis and tables and findings are limited to those patients (n=647) who had one or more symptoms and who were not diagnosed by lung cancer screening.
Reviewer comment
It is noted that asymptomatic and screen-detected patients were excluded from the analysis from section 3.4 moving forward. However, it raises the question as to whether they should have been included in the study at all, even in terms of assessment of symptoms and signs prior to diagnosis, as this should be a different patient population than non-screen detected cancers.
Author response
It is quite reasonable, as noted above, to describe what proportion of our cohort of individuals with lung cancer were diagnosed by screening (a small minority) vs after symptomatic presentation. This finding further emphasizes the point that screening remains an infrequent route to diagnosis for most patients with lung cancer.
Reviewer comment
Having said that, while screen detected patients are excluded for the rest of the analysis, it would be more important to include these patients in the analysis of time from abnormal imaging to diagnosis and time to treatment. Once a patient has an abnormal CT scan, the pathways for screen detected and non-screen detected patients should proceed similarly. It’s the interval from symptoms to diagnosis that would be most impacted by screen-detected vs. non-screen detected differences.
Author response
We did not set out to compare timeliness from CT imaging to diagnosis for patients detected by screening, vs those detected after symptomatic presentation. This would be an interesting analysis for future studies with larger sample sizes
Reviewer comment
2.2 – Regarding symptoms and signs prior to diagnosis:
What percentage of patients with the listed symptoms in section 2.5 did not have lung cancer? What is the positive predictive value of the symptoms listed in this study?
Please explain why individuals with early-stage disease were more likely to complain of bone pain – this does not make sense clinically as early-stage patients wouldn’t have bony disease from lung cancer. As such, is it relevant as a sign/symptom if it does not correlate with lung cancer?
Author response
This study does not compare symptom/sign frequency between individuals with lung cancer vs controls without lung cancer. We have a separate manuscript currently under review (but cited here reference 17) which provides this data, and refer the Reviewer to this citation. Prado et al (ref 17) provides information on the odds ratios of multiple clinical features extracted using NLP from cases of lung cancer and matched controls.
In terms of bone pain (Appendix E), the Reviewer assumes that bone pain is due to metastases, which may not be a correct assumption. The direction of the association of lymphadenopathy with late-stage cancer on the other hand, does fit with what might be expected from later stage cancer. The combination of symptoms and signs (as well as additional data from changes in body weight, or changes in lab values etc.) that best predicts early vs later stage cancer in ambulatory care is outside of the scope of this article, but of high importance for future research.
Reviewer comment
Line 225 – Appendix D does not accompany the paper review materials
Author response
We are sorry you did not have the opportunity to read the appendices, they should be available now.
Reviewer comment
I don’t think Table 2 adds anything beyond what Figure 2 shows.
Author response
Thank you for your comment, and we would be happy to remove one of these, if the Editor agrees.
Reviewer comment
This paper is also limited in its lack of evaluation of outcomes related to timely care. What is the impact of delays in care on survival outcomes for various stages of disease? How do we know that faster care is better if the data shows the effects of “sicker quicker” care i.e. that the most symptomatic patients receive the fastest care?
Author response
We did not attempt to determine associations between time to diagnosis and outcomes (e.g. stage at diagnosis, survival). Indeed, there are remaining questions about what impact more rapid diagnosis after symptom onset would have on these outcomes, these are outside of the scope of this particular study.
Reviewer comment
Discussion
Rather than frame this study as informing metrics that could be used to develop a CQM that would measure the timeliness of cancer diagnosis, I think it would better to present this as a paper that describes a single center experience in timeliness of care for ambulatory lung cancer patients. It is not a fulsome enough evaluation to be able to inform a CQM around timeliness of care.
Author response
As noted in the above responses, we have clarified that this is a single center experience describing the pre diagnosis time course of lung cancer diagnosis, and using NLP which has never before been applied to ambulatory care records of patients with lung cancer to identify symptoms and signs. We fully appreciate that considerable further research will be needed to specificy a CQM that would be implementable in routine practice
Reviewer comment
This study does not add a lot to the literature – it is known that diagnostic intervals for lung cancer are too long, that the sicker-quicker effect exists whereby patients with advanced, urgent presentations receive faster care, and that many patients present with late-stage disease. What this study shows is the utility in local quality assurance processes of measuring timeliness of care, but it cannot inform system wide improvement strategies given the limited scope.
Author response
We strongly disagree. There have been no studies that we are aware of that have used routine EHR data to describe the pre-diagnosis time course of lung cancer in the USA. There have been several studies using claims data (e.g. , reference 9) or using coded symptom data in the UK (e.g. reference 8), some of which describe considerable periods in the pre diagnosis phase of lung cancer. Moreover no studies that we are aware of have applied NLP to the free text records of individuals with lung cancer to identify symptoms and signs from free text records in any country that we are aware of.
Reviewer comment
Line 300 “However, until there is clear evidence for an association between pre-diagnosis phase and stage of cancer, implementing a CQM with the sole intention of promoting a shift to earlier stage at diagnosis may be premature”. I suggest adding that “until there is clear evidence for an association between pre-diagnosis phase, stage of cancer, and lung cancer outcomes”. We know that earlier stage disease is associated with better outcomes, but we don’t know the impact of timeliness of care on survival – i.e. the extent to which delays in care impact survival, for which stages of disease, and what constitutes a significant delay likely to impact survival.
Author response
We agree and have modified this sentence to read as follows: However, until there is clear evidence for an association between pre-diagnosis phase, stage of cancer and lung cancer outcomes, implementing a CQM with the sole intention of promoting a shift to earlier stage at diagnosis may be premature.
Reviewer comment
Line 310 – again the authors reference a Danish study when referring to WHO targets and Canadian guidelines
Author response
Thank you for pointing out this error in citation, which we have corrected.
Reviewer comment
Line 325 – “This important finding suggests unrealized value in considering CQMs here”. I find it challenging to understand how non-specific symptoms such as cough and fatigue can be used to guide a CQM for lung cancer. Even more so, the fact that symptoms like bone pain were more common in early-stage disease (without bony metastatic disease by definition) seem to suggest that some symptoms may be unreliable in terms of facilitating a diagnosis. Is the time intervals from non-specific symptoms to diagnosis really the best quality measure to guide lung cancer care?
Author response
This is an interesting question, and we had considered multiple different methods for defining ‘when cancer could have started’. There is no perfect approach here, and almost no way of knowing with perfect certainty exactly the time point in a patient’s life at which symptoms (single, or combination) of possible cancer diverged from the symptoms that most individuals experience in normal life. However, we were guided by statements and reference from the existing literature (e.g. the Aarhus statement noted in reference 23) in terms of how we defined the ‘starting point’ of cancer onset. We present several definitions of the potential onset of cancer in Table 2 showing the impact of more stringent definitions (presence of one or more, to presence of 5 or more symptoms) and time to diagnosis. But, we should not we avoid referring to this as the onset of cancer, as we cannot know that.
Our sister paper (Prado et al, reference 17) as well as other papers cited (e.g. 8) do provide evidence that most patients with lung cancer present with somewhat non-specific symptoms, and a minority present with so-called red flag symptoms. This is the reality of the clinical presentation. As noted in Section 4.4, we anticipate that combining multiple data sources from the EHR each of which may provide relatively non-specific indications of cancer, together may provide sufficiently strong evidence of a signal to direct the clinician to take next steps, such as imaging studies. ‘
Reviewer comment
Line 226 – “The short duration we observed may be skewed by patients diagnosed at the time of surgery, rather than a measure of health system performance overall.” This is a methodological flaw in the analysis, and one of the reasons these types of analyses are so difficult. In the case of patients going to surgery, one strategy is to consider the date of decision to operate as a date of diagnosis, to better understand the time intervals. It would also be important to consider timeliness of care measures by lung cancer stage to help sort out to what extent the early-stage group diagnosed at surgery impacted this measure vs. the sicker quicker effect whereby patients with more advance and more symptomatic disease received faster care.
Author response
To be consistent, we kept the date of diagnosis as the date of pathological confirmation. Parsing out this issue in stages further reduces our goal to describe a pathway to diagnosis.
Reviewer comment
From a QI perspective, the better CQM measures would be time from abnormal imaging to diagnosis (either rule in lung cancer or rule out) – not just time to diagnosis of lung cancer patients alone. We also need to consider the value of other quality metrics beyond timeliness of care (efficiency of resource utilization, % patients receiving guideline recommended (effective) care, equitability of care). I think the paper needs to include more information in the discussion about how a) timeliness of care measures are challenging for lung cancer care given the many challenges identified in this study, b) we should consider to what extent there are other measures of quality of lung cancer care that should be monitored.
Author response
We agree there are many more issues to consider in lung cancer detection, and we cover most of the above points already in Section 4.4, including need for validation, interventional studies, demonstrating impact on outcomes, unintended consequences. To be concise and focus on our data, we opted to exclude other important issues that are beyond the scope of this paper.
Reviewer comment
Line 387 – “However, we found few differences in symptoms/signs with stage at diagnosis”. Is it not concerning that the study did not find that symptoms correlated with stage of disease – it makes one wonder whether the symptoms were associated with lung cancer at all.
Author response
Thank you for this thought-provoking idea. Our interpretation is that symptoms are present throughout lung cancer stages, providing more evidence that lung cancer is not a “silent” disease. As noted in above responses, both Prado et al (reference 17) as well as other papers in the primary care field (e.g. reference 8) provide evidence for the association with various symptoms and cancer
Reviewer comment
Conclusion line 395 – “It is surprising that the US has no widely accepted CQM for the diagnosis of lung 395 cancer, despite” – Is it surprising? This study shows the challenges with regards to establishing timeliness of care metrics, particularly in terms of the challenges in obtaining data, determine time intervals, and sorting out the “sicker quicker” effect.
Author response
Given other countries such as UK, Denmark, Canada (and the WHO) have adopted CQMs for cancer, yes—it is surprising the US does not. The fact that other developed nations have managed to overcome some of the challenges of developing a CQM suggests problems are not insurmountable. This is more so given the burden that lung cancer places on patients and their families, and the costs (human and financial) of late-stage diagnosis.
Reviewer comment
399 – “and potential for improving outcomes”. – I think it is important to note that this study does not address outcomes at all.
Author response
Agreed—we feel we did not mislead any intention of assessing outcomes, yet rather posited potential in the future.

Round 2
Reviewer 1 Report
Dear authors:
Thanks for your answer. Anyway, since the diagnostic procedure nowadays is different to the diagnosis procedure several years ago and the aim of your work is to study the time needed for diagnosis, it should be included not just the basic CT scan, but all the tests a patient must undergo to be able to be operated. If not, it can not be studied the "diagnostic time", just the "CT scan time".
Author Response
Author response We should further clarify that date of diagnosis, was the date when the patient received their final, confirmed tissue diagnosis. This is the date of diagnosis in the SEER cancer registry, is based on pathology samples as a norm, it is not based on CT scan (or other imaging tests that may be used in diagnosis or staging). Our definition of date of diagnosis does not to be modified.
Based on input from our lung cancer clinical experts, CT chest is still used as an initial imaging test in all patients, some may have had a chest X ray performed prior to this (for example in primary care clinics), and many have additional chest imaging (e.g PET) used in addition to chest CT as part of their diagnostic work up and staging, as noted by the Reviewer.
Therefore, we have clarified in the text that chest imaging in the context of this paper, refers only to chest Xray or chest CT imaging, not the wider range of potential chest imaging that can be used. We have also added this sentence to the Discussion section on page 11 ‘We are also aware that imaging tests in addition to chest CT are used in the diagnostic and staging workup of individuals with suspected lung cancer.’
Reviewer 2 Report
The authors have made a reasonable effort to address the comments, particularly in terms of providing more balanced language regarding implications on quality measures.
Page 2, line 85 – “However, the significant potential opportunities to improve care in the US…” – I believe there is a word missing. ? despite the significant potential opportunities?
Author Response
Thank you for noting this, we have corrected the sentence as follows:
'However, despite the significant potential opportunities to improve care in the US [16], there are no CQMs focused on the timeliness of diagnosis of lung cancer currently used in the US.'
Round 3
Reviewer 1 Report
Dear authors; I understand you have followed the SEER cancer registry, but in my opinion, this work doesn't reflex a real diagnostic time study.